# Rare-event sampling analysis uncovers the fitness landscape of the genetic code

Yuji Omachi[1], Nen Saito [2,3,4] *, Chikara Furusawa [1,4,5] *

**1** Graduate School of Sciences, The University of Tokyo, Hongo, Tokyo, Japan, **2** Graduate School of Integrated Sciences for Life, Hiroshima University, Higashi-Hiroshima City, Hiroshima, Japan, **3** Exploratory Research Center on Life and Living Systems, National Institutes of Natural Sciences, Okazaki, Aichi, Japan, **4** Universal Biology Institute, The University of Tokyo, Hongo, Tokyo, Japan, **5** Center for Biosystems Dynamics Research, RIKEN, Suita, Osaka, Japan

☸ These authors contributed equally to this work.
* nensaito@hiroshima-u.ac.jp (NS); chikara.furusawa@riken.jp (CF)

**Data Availability Statement:** Codes and data are available at the ZENODO repository: https://zenodo.org/record/7663206#.Y_WxHuzP3SV.

**Funding:** This study was supported in part by the Japan Society for Promotion of Science (JSPS) KAKENHI (17H06389, 22K21344 to CF), Japan

## Abstract

The genetic code refers to a rule that maps 64 codons to 20 amino acids. Nearly all organisms, with few exceptions, share the same genetic code, the standard genetic code (SGC). While it remains unclear why this universal code has arisen and been maintained during evolution, it may have been preserved under selection pressure. Theoretical studies comparing the SGC and numerically created hypothetical random genetic codes have suggested that the SGC has been subject to strong selection pressure for being robust against translation errors. However, these prior studies have searched for random genetic codes in only a small subspace of the possible code space due to limitations in computation time. Thus, how the genetic code has evolved, and the characteristics of the genetic code fitness landscape, remain unclear. By applying multicanonical Monte Carlo, an efficient rare-event sampling method, we efficiently sampled random codes from a much broader random ensemble of genetic codes than in previous studies, estimating that only one out of every $10^{20}$ random codes is more robust than the SGC. This estimate is significantly smaller than the previous estimate, one in a million. We also characterized the fitness landscape of the genetic code that has four major fitness peaks, one of which includes the SGC. Furthermore, genetic algorithm analysis revealed that evolution under such a multi-peaked fitness landscape could be strongly biased toward a narrow peak, in an evolutionary path-dependent manner.

## Author summary

The fitness landscape is a conceptual geometric structure that describes the relationship between genotypes and fitness (i.e., reproductive success). The evolutionary dynamics of a population can be viewed as an adaptive walk on the landscape toward a higher fitness region. Thus, by analyzing the landscape, one can infer how evolution has shaped current organisms, or which genotypes have high fitness. Here, we analyzed the fitness landscape of the genetic code, the rule by which codons encode amino acid. The standard genetic

Science and Technology Agency (JST) ERATO (JPMJER1902 to CF), and Cooperative Study Program of Exploratory Research Center on Life and Living Systems (ExCELLS; program No. 20-102, 21-102 to NS). The funders had no role in study design, data collection and analysis, decision to publish, or preparation of the manuscript.

**Competing interests:** The authors have declared that no competing interests exist.

code (SGC), shared by almost all organisms, has an apparently non-random structure, implying that it has been subject to strong selection pressure. By applying multicanonical Monte Carlo, a numerical method developed in statistical physics, we visualized the genetic code fitness landscape. The landscape has four peaks, one of which contains the SGC, with the other three representing genetic codes that potentially could have been the current genetic code. Genetic algorithm analysis revealed that, in such a multimodal fitness landscape, evolution could be strongly biased toward narrower peaks, in an evolutionary pathway-dependent manner.

## 1 Introduction

The genetic code is the rule encoding the mapping between 64 codons and 20 different amino acids or a terminal codon. The standard genetic code (SGC), shared by almost all organisms, has an apparently non-random structure [1], in which the same amino acid is coded blockwise across adjacent codons in the codon table (Fig 1A; a block corresponds to a set of codons coding the identical amino acid), and is thus thought to be subject to selection pressure. Based on this, Woese [1] proposed the error-minimization hypothesis, which postulates that the SGC is selected to enhance robustness against translational (or possibly mutational) errors. Several subsequent theoretical studies, comparing the SGC to hypothetical random genetic codes in terms of robustness, support this hypothesis [2–6]. These studies estimated robustness using a simple formula [2–7] that can apply to any given genetic code, and the estimated robustness

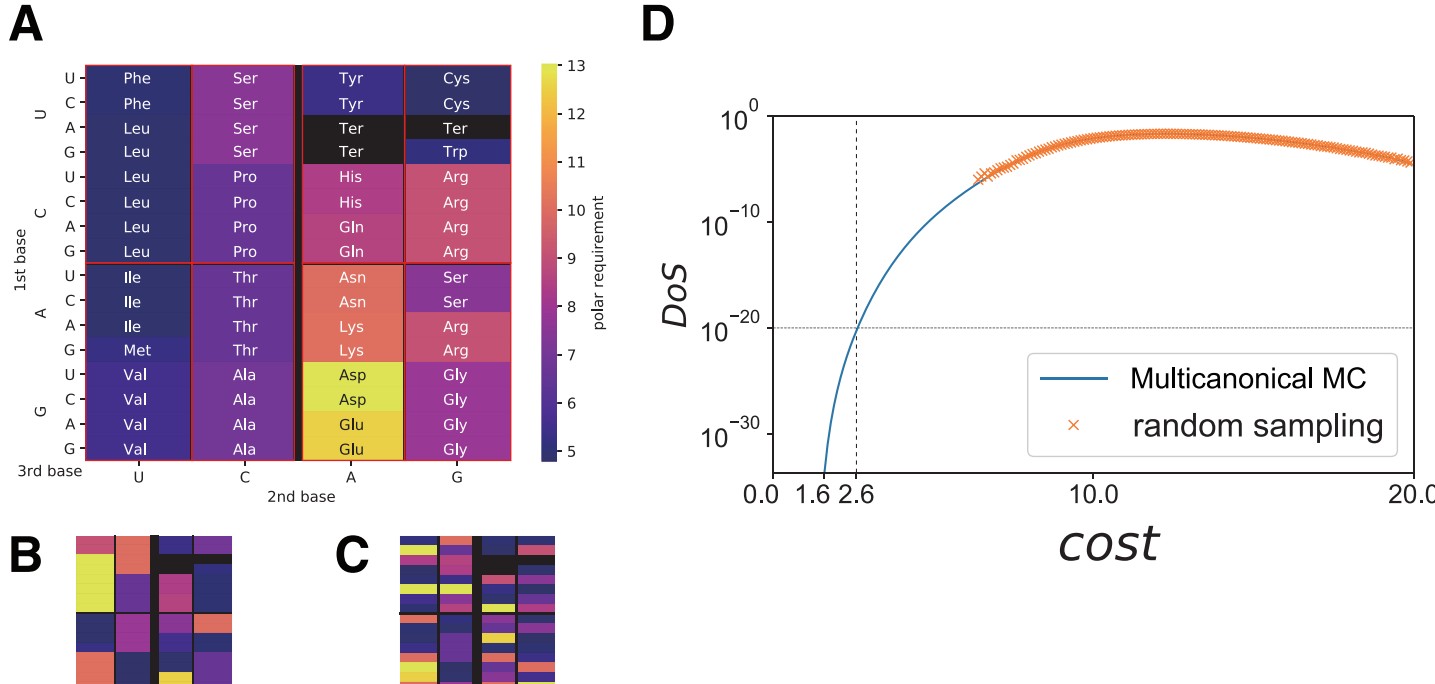

**Fig 1. The standard genetic code and randomly generated genetic codes.** (A) The standard genetic code. (B) A random genetic code generated from a random ensemble that Haig and Hurst [2] adopted. The block structure in the SGC where the same amino acid is coded is constrained to be maintained in the randomly generated code (see also S2 Fig). (C) A random code generated from a fully random ensemble where each codon maps to any one of 20 amino acids. (D) Cost density-of-states function for the proposed random genetic code ensemble. The blue line represents the estimates by the multicanonical Monte Carlo (multicanonical MC). The orange crosses indicate estimates based on naive random sampling.

can be regarded as the genetic code fitness. By estimating the fitness for multiple possible codes, this formula in principle enables analysis of the fitness landscape of the genetic code. The fitness landscape is a useful concept to describe evolutionary dynamics since evolution can be viewed as an adaptive walk on the landscape toward a higher-fitness region. The analysis of the fitness landscape of the genetic code would thus provide a lot of information about how the genetic code has evolved to the SGC and whether the evolution to the current SGC was inevitable, or whether it could have evolved to a completely different genetic code. Two different approaches have been used to analyze the fitness landscape of the genetic code: The first is a random sampling-based approach, which has shown how unlikely a code with a high fitness comparable to SGC appears by chance [2–6, 8, 9], or the existence of a huge number of local fitness peaks [8], etc. The second is an optimization-based approach. For example, studies using evolutionary algorithms have also demonstrated that the SGC is not at the global optimum [10–13]. However, these studies have been restricted to exploring a rather limited repertory of genetic codes due to high computational cost and thus analyzing only the local structure of the landscape. Because of this, the code evolution to the current SGC has not yet been clarified, as well as whether the high fitness codes all have a similar structure to the SGC or can have a completely different structure from the SGC. To elucidate these, the global structure of the fitness landscape needs to be analyzed. A single prominent peak with many small local peaks in the global structure of the fitness landscape would lead one to conclude that all high-fitness codes have a structure similar to that of the SGC. On the other hand, if there are multiple prominent peaks, each with many local peaks, it means that the current SGC occurrence is not a unique solution, but contains some coincidence.

A genetic code can be represented by a table with 64 entries, in which each entry corresponds to a single codon, a sequence of three DNA bases (A, G, C, T) or RNA bases (A, G, C, U). The genetic code maps each codon to one of 20 different amino acids or to a terminal signal (Fig 1A). Here, the code $a$ is a 64-dimensional vector, where each entry $a(c)$ for $c = 1$ ($UUU$), . . ., 64($GGG$) takes one of 20 different amino acids or a terminal signal. If a translation error causes codon $c$ to be mistakenly recognized as $c'$, an incorrect amino acid $a(c')$ is introduced into the protein. This error would cause an associated fitness reduction due to protein misfolding or malfunction. It is natural to assume that the reduction in fitness depends on the difference in physicochemical properties $d(a(c), a(c'))$ between amino acids $a(c)$ and $a(c')$. Given that hydrophobic interaction is the dominant driving force in protein folding, this difference in physicochemical properties can reasonably be represented as a difference in hydrophobicity or hydrophilicity. Indeed, Haig and Hurst [2] examined four amino acid's properties (polar requirement, hydropathy, molecular volume, and isoelectric point) and suggested that the SGC has been under selection pressure to minimize changes due to misreading in the polar requirement scale [14], a measure associated with hydrophilicity. Following these studies, we adopted $d(a(c), a(c'))$ as the square of the difference in the polar-requirement scale between $a(c)$ and $a(c')$. The polar-requirement values of each amino acid are illustrated by the colors in Fig 1A, and described in S1 Text. The cost of misreading a codon associated with code $a$ is given by the average $d(a(c), a(c'))$ under the probability $P(c'|c)$ of misreading $c$ as $c'$ (see Method section).

The calculation of the cost (see Eq 1 in Method) for randomly generated codes provides insights into how evolutionarily optimized the SGC is, in terms of robustness against translational error, and how unlikely it is that the current SGC evolved by coincidence (i.e., without selection). In their seminal paper, Freeland and Hurst [3] generated random genetic codes using a random sampling method, estimated the fraction of codes with lower costs (i.e., higher robustness against translational error) than the SGC, and concluded that the probability of obtaining equally or more robust genetic code is $\sim 10^{-6}$. Although these studies provide an

excellent basis for understanding genetic code evolution, they were limited in their search space due to the high computational cost and considered only a random ensemble of codes maintaining the codon block-structure of the SGC, in which a block corresponds to a set of codons coding the identical amino acid (Fig 1B and S2 Fig). Most of the studies in this field have considered similar random genetic code ensembles using limited search spaces. Such studies have examined, for instance, evolutionary simulation of the genetic code [8], and methods to improve the cost function to incorporate protein amino-acid frequency [4, 15]. However, this choice of random genetic code ensemble is inconsistent with examples of the non-standard genetic code [16–18], whereby the code differs in only a single or a few codons from the SGC, altering the block structure of the SGC. A justifiable choice of the random genetic code ensemble is such that each codon can take 20 possible amino acids, with the constraint that the whole code contains 20 different amino acids (Fig 1C); thus, the number of possible codes in the ensemble is approximately $20^{64} \sim 10^{83}$, which is much larger than the number of possible codes in the ensemble ($20! \sim 10^{18}$) maintaining the same block structure of the SGC. However, a fully random ensemble with $20^{64}$ states produces a 64-dimension search space. Given this size of the search space, naive random sampling, in which all random genetic codes are generated with equal probability, cannot be used to obtain a meaningful genetic code with a cost as low as that of the SGC.

Here, we applied multicanonical Monte Carlo (MC) analysis, which can effectively sample low-cost genetic codes even from an ensemble of fully random genetic codes, to reveal the global structure of the fitness landscape. The resulting fitness landscape had four major peaks, one of which includes the SGC. This indicates that there are four classes of high-fitness genetic code and suggests that the current genetic code potentially could have been the other three classes.

## Method

The average cost of misreading a codon associated with code $\boldsymbol{a}$ is given by

$$cost(\boldsymbol{a}) = \sum_c \sum_{c'} P(c'|c) d(a(c), a(c')),  \tag{1}$$

where $P(c'|c)$ is the probability of misreading $c$ as $c'$. For the computation of Eq (1), we adopted the following $P(c'|c)$ according to previous studies [3–6, 15] that are based on experimental observation [19]:

$$P(c'|c) \propto \begin{cases} 1 & \text{for a change in the 3rd base} \\ 1 & \text{for a change in the 1st base by TS} \\ 0.5 & \text{for a change in the 1st base by TV} \\ 0.5 & \text{for a change in the 2nd base by TS} \\ 0.1 & \text{for a change in the 2nd base by TV} \\ 0 & \text{double/triple base changes} \end{cases},  \tag{2}$$

where TS represents a transition [an interchange between purines (A ↔ G) or between pyrimidines (C ↔ U)], and TV denotes a transversion [an interchange of purines for pyrimidines (A ↔ C, A ↔ U, G ↔ C, G ↔ U)]. This $P(c'|c)$ indicates that the most frequent errors are mistranslations within the red box in Fig 1A, while misreading of a codon into a codon in an adjacent red box, without crossing the black thick line at the center, is half as likely, and the misreading of a codon by crossing the center line is 10 times less likely (S1 Fig). For misrecognition of an amino acid as the terminal signal (where $a(c')$ is the terminal signal), a cost $c_0$,

independent of the polar requirement of $a(c)$, is assigned. Likewise, for misreading the terminal signal as an amino acid (that is, $a(c)$ is the terminal signal) a constant cost $c_1$ is assigned. As the choice of $c_0$ and $c_1$ affects only the basal value of the cost, and does not alter the cost difference between any two codes, we adopted $c_0 = c_1 = 0$. Only a single base change is considered here, because the probability of double or triple base changes are negligibly small.

Under this cost function Eq 1 with misreading probability Eq 2, we consider numerical construction of random genetic codes with high robustness against mistranslation. We hypothesized that the random codes are required to satisfy the following fundamental property of the SGC: (1) Each codon can encode one of 20 amino acids. (2) The terminal signals should be coded in the same position as the SGC, and their number is also the same as the SGC. (3) The code must encode the full repertoire of amino acids (i.e., 20 amino acids). (4) The number of aspartic acids (Asp) and glutamates (Glu) in the code should be two or more (i.e., at least the same number as the SGC). (1) is our original condition and differs significantly from previous studies [2–6, 8, 9] that assumed a block structure for the genetic code. Conditions (2) and (3) are required to avoid generating biologically meaningless genetic codes, such as those comprising only the terminal signal, which would obviously have the lowest cost. Further, we here required an additional condition (4) because Asp and Glu have the highest polar-requirement values, and the cost value depends critically on their numbers. Without this requirement, almost all ($\sim$ 99%) of the sampled genetic codes with low cost contain only a single Asp and Glu, which is not compatible with our goal of visualizing the fitness landscape to which the SGC belongs (see S1 Text and S5 Fig). This restriction enforcing to have two Asp and two Glu each does not arise naturally from considerations of code robustness but is rather artificial. As is discussed below in Results section, we confirmed that our main claims do not depend on this condition, while the structure of a low-cost random code was dissimilar to the SGC under the absence of the condition.

For sampling from such an ensemble, we apply a computational technique termed as rare event sampling using multicanonical Monte Carlo [20, 21]. Multicanonical Monte Carlo has been developed in statistical physics such as spin systems [20–22] and protein folding [23, 24]. This method enables efficient sampling of rare events, exemplified by low-energy states, without becoming trapped in local minima. Furthermore, this method provides an estimate of the density-of-state, which represents the energy distribution for all possible states in the system. This method combines the advantages of an optimization algorithm (energy minimization) and naive random sampling (calculation of energy distribution), and has been applied to a variety of fields other than physics, such as random matrix theory [25], combinatorial optimization [26], and the generation of surrogate time-series data [27]. Recently, this method has also been applied to gene regulatory network evolution [28–30].

Here, we consider the Markov chain sampling of random genetic code to be $\boldsymbol{a} \rightarrow \boldsymbol{a}' \rightarrow \ldots$, where a candidate state $\boldsymbol{a}'$ is generated from $\boldsymbol{a}$ by flipping the $c$-th index of $\boldsymbol{a}$, $a(c)$, to another amino acid $a'(c)$ [$\neq a(c)$]. The transition $\boldsymbol{a} \rightarrow \boldsymbol{a}'$ is determined by accepting or rejecting the candidate state $\boldsymbol{a}'$ of the transition probability function, $\Pi_{cost(\boldsymbol{a}) \rightarrow cost(\boldsymbol{a}')} = \min(1, w(cost(\boldsymbol{a}'))/w(cost(\boldsymbol{a})))$, where $w$ is the "weight" function (a function of the cost). The sampled distribution of the cost, $P_{eq}(cost)$, obeys the following detailed balance condition for a large sample size:

$$\Pi_{cost \rightarrow cost'} P_{eq}(cost) = \Pi_{cost' \rightarrow cost} P_{eq}(cost'), \tag{3}$$

and thus depends crucially on the weight. While the Gibbs distribution of $P_{eq}(cost)$ is obtained for the choice of an exponential function of $cost$ in ordinary Metropolis sampling, multicanonical sampling adopts the multicanonical weight $w(cost) \propto 1/\Omega(cost)$, where $\Omega(cost)$ is the

density-of-state, leading to a uniform distribution of $P_{eq}(cost)$, since $P_{eq}(cost) \propto \Sigma_{\boldsymbol{a}} \, w(cost(\boldsymbol{a}))I(cost(\boldsymbol{a}) = cost) = w(cost)\Omega(cost)$, where $I(X = Y)$ is the function that takes $I(X = Y) = 1$ if the equality is satisfied, and zero otherwise. Therefore, for multicanonical sampling, the sequence of states $\boldsymbol{a}$ generated by the Markov chain is regarded as a random walk in cost space, ensuring efficient sampling of rare events (i.e., low cost states) without becoming trapped in local minima. Since $\Omega(cost)$ is not known a priori, one needs to numerically compose $w(cost)$ that gives rise to a flat sampling histogram of the cost space. Here, we used the Wang-Landau algorithm [22] to construct and tune the weight function $w(cost)$, subsequently performing multicanonical sampling with a fixed $w(cost)$ [27].

We also performed a simple genetic algorithm with population size $N = 100$. Initially, randomly generated $N$ genetic code that satisfies the above (1)-(4) conditions are prepared. In each generation, the top 50% of the population of genetic codes was selected according to the cost in Eq (1) and copies of the selected population are generated by random sampling without replacement until the total population reaches $N = 100$. In these copies, mutations in the genetic code (i.e., from $a(c)$ to another amino acid, $a'(c)$) occur with a mutation rate $\mu = 0.1$ in each element of $\boldsymbol{a}$. The recombination (crossover) process [10–13] is not incorporated. When the mutated code does not contain 20 different amino acids, or at least two instances of Asp and Glu, the mutation process starts over. We performed $10^5$ independent GA runs with 500 generations and analyzed the statistics of the trajectories.

## 2 Results

First, we computed the fraction of genetic codes having a lower cost than the SGC in a random genetic code ensemble. The total number of possible genetic codes in the proposed ensemble was $\sim 10^{79}$, which is still much larger than that reported in previous studies [3–6, 8, 15]. Naive random sampling with a sample size $10^{10}$ could not generate a code with a lower cost than the SGC (Fig 1D). We performed multicanonical sampling with $10^{11}$ steps by fixing the weight function $w(cost)$. The density of states $\Omega(cost)$ is obtained by $\Omega(cost) \propto H(cost)/w(cost)$, where $H(cost)$ is the sampling histogram of the cost. The estimates of $\Omega(cost)$ (Fig 1D) were consistent with the naive random sampling results for a large cost, revealing that the density of genetic codes having a cost lower than that of the SGC is ca. $10^{-20}$. Such a genetic code is virtually impossible to generate via naive random sampling. This estimate is much smaller than the previous estimates, $\sim 10^{-6}$, indicating that the SGC is much more evolutionarily optimized for robustness against errors than previously thought.

Next, we investigated the characteristics of genetic codes with low cost (i.e., high fitness). Using the sampled data with costs comparable to, or lower than, those of the SGC (i.e., $cost < 2.6 + \Delta$, where $\Delta = 0.1$), we reduced the 64-dimensional code space to two-dimensional space, via principal component analysis (PCA). This use of a subset of random codes for PCA is based on our assumption that only low-cost random codes can form a meaningful low-dimensional structure. PCA was performed for the set of 64-dimensional vectors $\boldsymbol{a}$, in which each entry contains the polar-requirement scale of the encoded amino acid at the codon. The resultant scatter plot in the PC1-PC2 space for $cost = 2.6 \pm \Delta$ (the same cost as the SGC) (Fig 2A) reveals that there are four clusters, among which the data are unevenly distributed ($\sim 44\%$ in the red cluster, $\sim 31\%$ in the orange, $\sim 23\%$ in the green, and $\sim 3\%$ in the blue). No further structures were found in the PC1-PC2-PC3 space (S3 Fig). To determine the number of clusters, we used the k-means method with the elbow method (S3 Fig). By piling up a two-dimensional scatterplot of different cost values on the vertical axis, we can visualize the cost landscape, the counterpart of the fitness landscape, where low cost corresponds to high fitness,

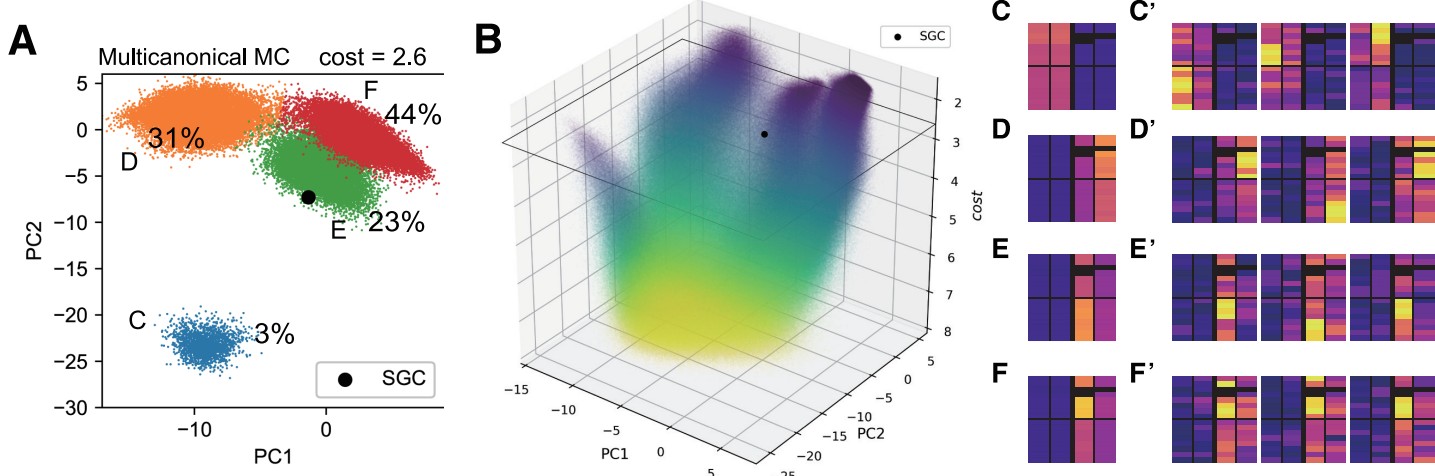

**Fig 2. Fitness landscape under a random genetic code ensemble.** (A) Genetic codes with comparable cost values to the SGC, where $2.6 - \Delta < cost < 2.6 + \Delta$ ($\Delta = 0.1$) in the PC1-PC2 space (the contribution ratios: PC1, 27%, PC2, 14%). Principal component analysis and k-means clustering were performed using the sampled data with a cost value of $cost < 2.6 + \Delta$. (B) Fitness landscape visualized by piling up the scatterplot on the PC1-PC2 plane, with costs decreasing on the vertical axis, where low cost corresponds to high fitness. (C-F) Average structure of the genetic code in each cluster. (C'-F') examples of samples belonging to the clusters (C-F) in (A).

and vice versa. Fig 2B clearly shows the multimodality of the fitness landscape, where the four-peak structures in the low-cost region are connected to the high-cost region. Fig 2C–2F illustrate the average composition of the genetic codes within each peak for a genetic code with the same cost as the SGC ($cost = 2.6$), and Fig 2C'–2F' show three representative examples in each cluster. Comparing Fig 2A and 2C–2F, we see that a genetic code with high PC1 values tends to have high polar-requirement amino acids in the third column, whereas the larger the value on PC2, the more high polar-requirement amino acids tend to be coded in the third and fourth columns (the right half of the codon table). The green cluster in Fig 2A, to which the SGC belongs, was further analyzed via PCA of the dataset belonging to the cluster. This revealed a similar genetic code structure to the SGC (S4 Fig).

So far, we considered an ensemble containing at least two Asp and two Glu (i.e., condition (4) in the Methods section). By performing additional simulations, we also confirmed that our results do not depend on this condition. A closely similar fitness landscape with four peaks appears even in the absence of the condition (see S1 Text and S5 Fig). Further, we performed simulations with two different cost (fitness) functions: (1) the cost including amino acid frequencies in the genome in extant organisms, and (2) the cost incorporating an empirical evolution-based scoring function for amino acid substitution (BLOSUM62 score [31]). In both cases, similar fitness landscapes with multi-peaks were confirmed (S6 and S7 Figs). In cost incorporating BLOSUM score, the orange cluster in the original model (Fig 2: the cluster (d)) that codes high polar requirement amino acids at the fourth column is split into two clusters (d) and (f) under this cost function (S7 Fig; the orange and purple clusters in (A)). However, the amino acid configuration for each peak is similar to the original model (Fig 2C–2F).

Our use of multicanonical MC enabled unbiased sampling of random genetic codes, enabling us to analyze and visualize the global structure of the genetic code fitness landscape. In the reconstructed fitness landscape, the optimized codes were categorized into four clusters (Fig 2A and 2B): the blue cluster, in which highly polar amino acids are located in the two left columns, occupies only a small fraction ($\sim 3\%$, for $cost < 2.6 + \Delta$). Interestingly, genetic algorithms (GAs), widely used evolutionary simulations, can cause large deviations from unbiased sampling, because the evolutionary outcome could be strongly biased in an evolutionary path-

dependent manner. To demonstrate this, we performed a GA under the cost in Eq (1) (see details in Method section).

Fig 3A illustrates the resultant genetic codes that reached a cost of 2.6 (the same cost as the SGC) on the same PCA projection as in Fig 2A; this further illustrates that the GA can reach all four peaks, with up to 30% of the GA results concentrated in the blue cluster. However, the actual phase volume of the cluster occupies only 3% of the high-fitness code space (Fig 2A, blue cluster). To understand how this bias arises, we analyzed the optimization trajectories of the independent GA runs (Fig 3B–3E). The blue cluster in Fig 3A comprises codes where high polar-requirement amino acids are located in the two left columns (see also Fig 2C and 2C'); the tendency to approach the cluster via a GA trajectory is characterized by $\Delta_{PR}$, the difference between the average polar-requirement value in the two right columns $\langle PR \rangle_R$ and the two left columns $\langle PR \rangle_L$ of the code. Fig 3D illustrates that $\Delta_{PR} = \langle PR \rangle_R - \langle PR \rangle_L$ through GA

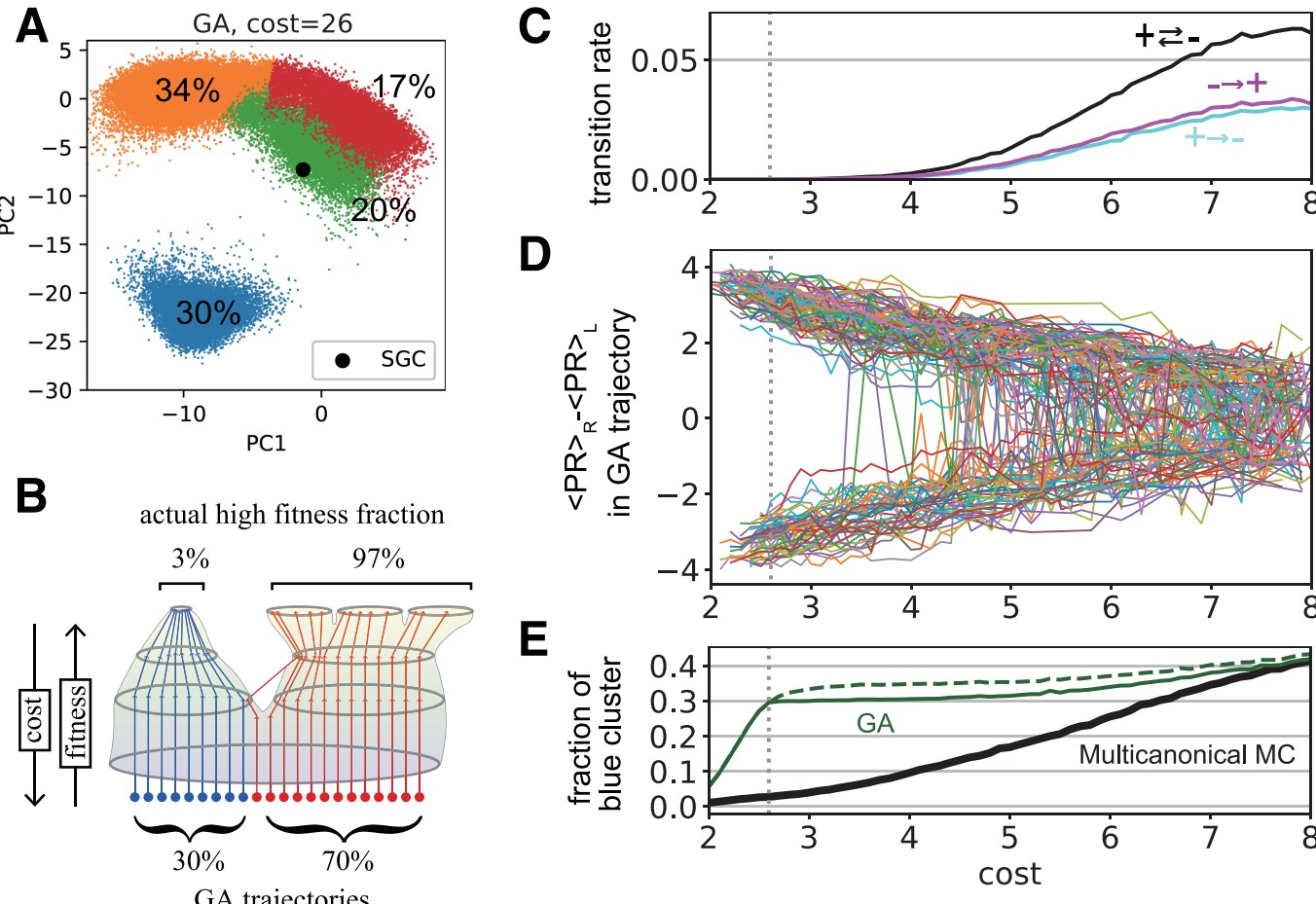

**Fig 3. Simulation results by genetic algorithms (GAs).** Results of $10^5$ independent GAs. (A) Codes with comparable cost values to the SGC, $2.6 - \Delta < cost < 2.6 + \Delta$ ($\Delta = 0.1$), on the same PC1-PC2 plane as in Fig 2. Clustering was performed based on samples with $cost < 2.6 + \Delta$, obtained via GA and MCMC sampling. (B) Schematic illustration of evolution under the multipeak fitness landscape. (C-E) Analysis of the independent GA trajectories. The cost values of the independent GA runs are plotted on the horizontal axis. (C) Transition rates from $\Delta_{PR} > 0$ to $\Delta_{PR} < 0$ (cyan), from $\Delta_{PR} < 0$ to $\Delta_{PR} > 0$ (purple), and between $\Delta_{PR} > 0$ and $\Delta_{PR} < 0$ (black), where $\Delta_{PR}$ is the difference in polar requirement between left and right two columns of code: $\Delta_{PR} = \langle PR \rangle_R - \langle PR \rangle_L$. (D)$\Delta_{PR}$ in GA trajectories as a function of the cost value experienced in each GA. Note that if a trajectory experiences the same cost value multiple times, only the latest value is recorded, so a single curve is always a univalent function and does not necessarily correspond to the GA time series (GA time series can go up and down in cost). (E) Fraction of genetic codes with $\Delta_{PR} < 0$. The green dashed line depicts the fraction for all the GA trajectories, while the thick line indicates that for GA trajectories conditioned to reach cost = 2.6.

trajectories as a function of the cost values experienced in a single GA run, where genetic codes with negative $\Delta_{PR}$ values around *cost* = 2.6 represent the blue cluster in Fig 3A. In Fig 3C, the number of transitions between positive and negative $\Delta_{PR}$ declines as the optimization proceeds, with almost none occurring below *cost* = 4 ∼ 5. Consistent with this, the fraction of genetic codes with $\Delta_{PR} < 0$ appearing in the GA optimization trajectories exhibits a plateau below *cost* = 4 ∼ 5 (Fig 3E: thick green line), whereas the actual fraction of codes estimated via multicanonical MC (Fig 3E: thick black line) declines monotonically as cost decreases. This indicates that the fraction of codes with a negative $\Delta_{PR}$ below *cost* ≃ 2.6 (i.e., the blue cluster in Fig 3A) is determined early in the optimization process, at *cost* > 5, being determined by the phase-space volume with *cost* > 5 rather than *cost* ≃ 2.6. This leads us to conclude that the GA can cause a large bias in the situation depicted in Fig 3B, where the fitness landscape has multiple peaks that allow no peak-to-peak jump, and one peak (the left peak in Fig 3B) has a broad waist and narrow tip. In such a situation, the fraction of trajectories reaching the left peak is determined by the waist size of the fitness landscape; in other words, the basin size of the peak, when the optimization dynamics are interpreted as a dynamic system, causes the overconcentration in the left narrow peak.

## 3 Discussion

The proposed multicanonical Monte Carlo method, which efficiently samples rare events in a fully random genetic code ensemble, revealed that the probability of obtaining a genetic code with a cost as low as that of the SGC by chance is ca. $10^{-20}$, much lower than the previous estimate [3]. It would be virtually impossible to generate such a code by naive random sampling. The proposed multicanonical method would therefore be appropriate as a standard tool in studies inferring the selection pressure on the SGC, instead of the previously used naive random sampling method [3–8, 15]. Note that despite their extremely low probability, the number of such codes was huge as ∼ $10^{59}$. This "numerous but rare" problem is also found in the "magic square" problem [26].

The genetic code fitness landscape had four peaks, one of which included the SGC, indicating that the genetic code might have evolved as one of the other three classes. We also performed GA simulations to analyze possible evolutionary trajectories toward these peaks. Comparing the GA trajectories and the fitness landscape constructed through Monte Carlo sampling revealed significant differences in the distribution of low-cost codes (Figs 2A and 3A), which represents that the evolutionary path-dependent bias in the GA arises under the multi-peaked fitness landscape (e.g., Fig 3B). Specifically, the fraction of independently evolved genetic codes reaching each peak is determined by the basin volume rather than the phase-space volume of these optima. This finding demonstrates the importance of integrating sampling-based and optimization-based approaches. From the perspective of statistical physics, this fraction is dictated by the sum of the path probabilities from a random initial condition to each peak, rather than the Boltzmann distribution of the fitness function; the fraction should coincide with the Boltzmann distribution when the system is in equilibrium but can differ in a finite time. In the proposed scenario, the evolutionary outcome is determined by the ease of finding a peak of the fitness landscape, but not by the phase-space volume of optimal genotypes. This scenario is applicable to both evolutionary studies and optimization problems with gradient-descent-like algorithms under a multi-peaked fitness landscape. Evolutionary path-dependent bias has also been reported in a study of gene regulatory networks [29], in which the mutational robustness was significantly higher in the GA than in the reference ensemble at the same fitness level. It remains to be clarified whether this bias toward mutational robustness

[29, 32] is related to the bias caused by the presence of multiple peaks, presenting a new avenue for evolutionary studies.

The asymmetric four-peak shape of the fitness landscape (Fig 2B) is attributed to our assumption that the evolution of the terminal signal is ignored. However, this does not mean that the results obtained are mere artifacts with no meaning to interpret, nor that our assumptions are unrealistic. An asymmetric fitness landscape (e.g., Fig 2B) emerges naturally when the terminal codons are localized on the table to reduce the cost, as in the case of SGC. In this sense, the fitness landscape obtained in this study should be considered as a landscape under the conditional probability that the terminal signals are in the same position as in the SGC.

We also examined an ensemble without the condition that two Alu or Glu must be included (S5 Fig). Further, the additional simulations were performed with two different cost functions: a cost incorporating amino acid frequencies in genome data (S6 Fig) and a cost based on BLOSUM62 score [31] (S7 Fig). In each case, similar fitness landscapes with multiple peaks were identified (S5–S7 Figs; see also S1 Text). From these results, we concluded the presence of four (or five) major peaks in the fitness landscape of the genetic code. The following aspects that were beyond the scope of our study require future analysis: the incorporation of the effect of codon usage [6, 15], coevolution of the genetic code with the amino-acid synthesis metabolic pathway [9, 33, 34], code robustness against frameshift[35–37], and the recently reported but controversial claim that the genetic code is robust against mutations that increase the uptake of carbon and nitrogen incorporation [7, 38, 39]. In particular, our analysis did not answer why nearly all organisms use the same genetic code, the SGC, instead of using different codes for different organisms. The emergence of a universal code could be explained by the effects of horizontal gene transfer (HGT), as Vetsigian et al. [40] addressed. How the HGT can alter the evolutionary outcome under the proposed fitness landscape will also be one of the issues to be addressed. Likewise, the present analysis provided no answer as to why the SGC stopped at a suboptimal location for a particular peak. Several hypotheses can be made: (a) the SGC has been frozen to the present position during evolution. (b) The code structure at a very early stage of evolution may have influenced the current SGC [9, 33, 34]. (c) We might fail to capture the true code fitness, where the SGC is the global optimal. Selection pressures other than robustness against mistranslations or amino-acid properties other than polar requirements may need to be considered. Our proposed method has a flexibility to incorporating several factors as listed above and would provide a promising means of estimating the rareness of SGC from a fully random genetic code ensemble, and visualizing the fitness landscape of the code.

## Supporting information

**S1 Text. The technical appendices include the details on (1) simulation with relaxing the requirement to have at least two instances of ASP and GLU, (2) simulation for examining the different cost functions that incorporate amino acid frequencies, and (3) a table of polar requirement scale for amino acids.**
(PDF)

**S1 Fig. Two examples of codon misreading.** (A) Example of GUG misreading. The star represents a codon to be misread by translational error. Blue, green, and red indicate erroneous changes at the 1st, 2nd, and 3rd bases, respectively. The number in the table indicates the weight for $P(c'|c)$, the probability of misreading $c$ as $c'$. For example, misreading is twice as likely when the weight is 1 as when it is 0.5. (B) Example of CCA misreading.
(PDF)

**S2 Fig. Random code ensemble used in previous studies.** (A) Procedure for generating a random code in the ensemble. The 20 different amino acids are randomly assigned numbers from (1) to (20), thus maintaining the SGC block structure that are groups of codons for where the same amino acid is coded. (B) The Standard genetic code (SGC). (C) Example of a random code generated by (A).
(PDF)

**S3 Fig. Cluster number determination for the scatter plot in Fig 2A.** (A) Visualization of the scatter plot Fig 2A in the PC1-PC2-PC3 space. (B) Elbow method for determining the number of the cluster = 4 in Fig 2A. The sum of square error (SSE) is plotted as a function of the number of clusters. The SSE decreases rapidly as the number of clusters increases from 1 to 4, then declines more slowly from 4 to 8 clusters.
(PDF)

**S4 Fig. PCA analysis of the dataset for the green cluster in Fig 2A.** (A) Scatterplot on the PC1-PC2 plane (PC scores: PC1, 31%; PC2, 7%). The symmetrical shape is attributed to the symmetry of the cost function, with respect to the swapping of the bottom four rows and the next four rows. (B-D) Examples of random genetic codes in the four corners in (A). The code in (C) is located near the SGC on the PC1-PC2 plane, and indeed has a similar structure to the SGC.
(PDF)

**S5 Fig. The fitness landscape when the requirement to have two instances of Asp and Glu is relaxed.** Here, we consider a random genetic code ensemble of genetic codes with 20 different amino acids (i.e., at least one instance of Asp and Glu). (A) Scatterplot of genetic codes with cost values comparable to those of the SGC $[2.6 - \Delta < cost < 2.6 + \Delta\ (\Delta = 0.1)]$ on the PC1-PC2 plane. (B) Fitness landscape visualized by piling up the scatterplot on the PC1-PC2 plane with different costs on the vertical axis. (C-F) Average structure of the random genetic code in each cluster. (C'-F') examples of samples belonging to the clusters (C-F) in (A).
(PDF)

**S6 Fig. Fitness landscape under a random genetic code ensemble incorporating the amino acid frequency in the genomes.** The cost function defined by Eq. S1 in S1 Text is used. (A) Genetic codes with comparable cost values to the SGC, where $2.9 - \Delta < cost < 2.9 + \Delta\ (\Delta = 0.1)$ in the PC1-PC2 space (left panel) and in the PC1-PC2-PC3 space (the contribution ratios: PC1, 31%, PC2, 18%, PC3 9%) (right panel). Principal component analysis and k-means clustering were performed using the sampled data with a cost value of $cost < 2.9 + \Delta$. (B) Fitness landscape visualized by piling up the scatterplot on the PC1-PC2 plane, with costs decreasing on the vertical axis (left panel). The peak that the SGC belongs to in the PC1-PC2 plane is also shown in the PC2-PC3 plane (right panel). (C-F) Average structure of the genetic code in each cluster. (C'-F') examples of samples belonging to the clusters (C-F) in (A). (G) Elbow method for determining the number of clusters = 4 in (A). The sum of square error (SSE) is plotted as a function of the number of clusters. (H) Cost density-of-states function for the ensemble estimated by the multicanonical Monte Carlo.
(EPS)

**S7 Fig. Fitness landscape under a random genetic code ensemble incorporating BLO-SUM62 score.** The cost function defined by Eq. S2 in S1 Text is used. (A) Genetic codes with comparable cost values to the SGC, where $3.3 - \Delta < cost < 3.3 + \Delta\ (\Delta = 0.1)$ in the PC1-PC2 space (the contribution ratios: PC1, 21%, PC2, 8%). Principal component analysis and k-means clustering were performed using the sampled data with a cost value of $cost < 3.3 + \Delta$.

(B) Fitness landscape visualized by piling up the scatterplot on the PC1-PC2 plane, with costs decreasing on the vertical axis. (C-G) Average structure of the genetic code in each cluster. (C'-G') examples of samples belonging to the clusters (C-G) in (A). (H) The sum of square error (SSE) in the elbow method is plotted as a function of the number of clusters. (I) Cost density-of-states function for the ensemble estimated by the multicanonical Monte Carlo. (EPS)

**S8 Fig. Amino acid frequency in random genetic codes from a random genetic code ensemble, where the genetic code contains 20 different amino acids with at least one instance of Asp and Glu.** (A) Percentage of genetic codes with one instance of Asp and Glu, against cost. At low costs, almost all of the genetic codes have only one instance of Asp and Glu. (B) Mean and variance of the number of amino acids in the random genetic codes, with cost $2.6 - \Delta < cost < 2.6 + \Delta$ ($\Delta = 0.1$), for the same ensemble as in (A). (C) The same plot as in (B), but for the random code ensemble with at least two instances of Asp and Glu in each genetic code. The numbers of amino acids in the SGC are within $2\sigma$ of the number obtained using the random ensemble. (PDF)

## Acknowledgments

We would like to acknowledge the helpful discussions with Yukito Iba, Macoto Kikuchi and Kunihiko Kaneko.

## Author Contributions

**Conceptualization:** Nen Saito, Chikara Furusawa.

**Formal analysis:** Nen Saito.

**Investigation:** Yuji Omachi, Nen Saito.

**Methodology:** Nen Saito.

**Project administration:** Nen Saito, Chikara Furusawa.

**Supervision:** Nen Saito, Chikara Furusawa.

**Validation:** Yuji Omachi, Nen Saito.

**Visualization:** Yuji Omachi, Nen Saito.

**Writing – original draft:** Nen Saito.

**Writing – review & editing:** Chikara Furusawa.

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
