## [Decision Letter · Decision Letter 0]

26 Oct 2022

Dear Dr Saito,

Thank you very much for submitting your manuscript "Rare-Event Sampling Analysis Uncovers the Fitness Landscape of the Genetic Code" for consideration at PLOS Computational Biology.

As with all papers reviewed by the journal, your manuscript was reviewed by members of the editorial board and by several independent reviewers. In light of the reviews (below this email), we would like to invite the resubmission of a significantly-revised version that takes into account the reviewers' comments.

All the reviewers found that the results presented in the paper under review are potentially interesting. They all, however, identified a number of shortcomings, which should be addressed before the study can be accepted for publication. I agree with the reviewers, and especially with the point related to considering a more general (and based on more recent data) cost function, moving beyond the hydrophobicity scale.

We cannot make any decision about publication until we have seen the revised manuscript and your response to the reviewers' comments. Your revised manuscript is also likely to be sent to reviewers for further evaluation.

Sincerely,

Artem Novozhilov

Guest Editor

PLOS Computational Biology

James O'Dwyer

Section Editor

PLOS Computational Biology

All the reviewers found that the results presented in the paper under review are potentially interesting. They all, however, identified a number of shortcomings, which should be addressed before the study can be accepted for publication. I agree with the reviewers, and especially with the point related to considering a more general (and based on more recent data) cost function, moving beyond the hydrophobicity scale.

Reviewer's Responses to Questions

**Comments to the Authors:**

Reviewer #1: Please find the review attached.

Reviewer #2: Omachi, Saito & Chikara Furusawa offer an estimate for the frequency of low-cost genetic codes using an advanced technique, multicanonical Monte Carlo sampling. They estimate that random genetic codes as robust as the Standard Genetic Code (SGC) with respect to hydrophobicity requirement of misread amino acids have frequency on the order of 10^-20; much lower than the previous estimates by Freeland & Hurst (10^-6).

Omachi et al. analysis is very interesting and certainly deserves to be published. There are, however, several issues that need to be addressed in the revision; at least at the level of the text editing, but, possibly, also by modification of the key expression in eq (1).

The core question here is what is a "random genetic code" (rGC)? The authors criticize Freeland & Hurst for retaining the block structure of SGC in their rGCs, noting that this is severely limiting the number of rGC choices. Instead they adopt a different way to generate block-like structure of robust rGCs, using the misreading probability (eq 2) that mimics the way SGC codons function in extant ribosomes. This "soft enforcement" is, arguably, a much better choice, but it should be clearly understood and explicitly stated that this is just a different choice of what SGC features are fundamental and which are dispensable.

Notably, while the authors initially introduce a very general definition of rGC (ll. 35-40), later they add some additional limitations (ll. 141-151) on the ground of being "compatible with our goal of visualizing the fitness landscape to which the SGC belongs". Conceptually, this is the same as enforcing the block structure, just a different choice of what SGC features are judged fundamental and what are deemed dispensable. Incidentally, I find that forcing the >=2 codon series for Asp and Glu is an unreasonable restriction in the context of this work; this restriction does not naturally arise from the hydrophobicity match requirements and if such rGCs are "better" than SGC, so be it.

I suggest to start the Methods section with a concise description of what the authors consider the fundamental properties of SGC that they require to be present in their rGCs (encoding the the full repertoire of AAs, with or without limitations on some of them, the specific form of misreading probability function, other limitations on terminators etc) and explain how their choices are different from those of their predecessors.

Besides the purely presentation level improvements, there are two ways that, I believe, could considerably improve the current analysis. First is going beyond the hydrophobicity requirement (that leads to solo-[DE] rGCs outperforming the SGC) and incorporating an empirical evolution-based scoring function for amino acid substitution cost (the easiest choice would be to incorporate alignment-derived log-odds matrices such as BLOSUM62 or PAM70). Second, assuming amino acid frequencies, mimicking those in the proteomes of extant organisms, would provide "soft enforcement" for the distribution of the size of the codon series (frequent AAs tend to use more codons), and also provide a way to meaningfully include the termination misreading cost that is ignored now (note how the SGC terminator codons are encoded in the same block with rarest amino acids). I realize that this would require a complete reanalysis, so I consider these requests to be optional (but highly recommended).

Reviewer #3: Reviewer's report of the manuscript “Rare-Event Sampling Analysis Uncovers the Fitness Landscape of the Genetic Code” by Yuji Omachi, Nen Saito, Chikara Furusawa

The manuscript presents results of applying multicanonical Monte Carlo method to sample efficiently random genetic codes in order to verify their robustness against translational errors. Using this method, the authors analysed much larger number of codes and found that there is only one code better optimized than the standard genetic code (SGC) out of 10^20. This estimation is smaller than the previous calculations, assuming one such code in a million. The authors characterized also the fitness landscape of the genetic code and found four major fitness peaks. In one of them was the SGC. The authors concluded that the SGC shows non-random structure and was subjected to strong selection pressure.

Although the application of this method is an advance in this field, the subject and the biological problem is not new because previous studied already conducted the random selection of codes but in much smaller number. The applied method is also not new because it was previously developed in statistical physics, protein folding and gene regulatory network evolution. Nevertheless, it was used to other problem. Although the study significantly increased the searched space of random codes, it is still a very small fraction in comparison to all possible codes, i.e. 10^83 or 10^59 depending on the assumption on the code structure.

Specific comments

The paper can be interesting, however, the authors ignored in Introduction studies that performed optimization of the SGC using various genetic and evolutionary algorithms. The studies should be cited in the background of current studies of the genetic code:

Błażej, P., Wnetrzak, M., Mackiewicz, P., 2016. The role of crossover operator in evolutionary-based approach to the problem of genetic code optimization. Biosystems 150, 61-72.

Błażej, P., Wnetrzak, M., Mackiewicz, D., Mackiewicz, P., 2018. Optimization of the standard genetic code according to three codon positions using an evolutionary algorithm. PLoS One, 13(8), e0201715.

Santos, J., Monteagudo, A., 2011. Simulated evolution applied to study the genetic code optimality using a model of codon reassignments. BMC Bioinformatics, 12, 56.

Wnetrzak, M., Błażej, P., Mackiewicz, D., Mackiewicz, P., 2018. The optimality of the standard genetic code assessed by an eight-objective evolutionary algorithm. BMC Evolutionary Biology 18, 192.

The studies showed that the SGC is not perfectly optimized as it could be expected from the adaptive hypothesis. It would be good to compare the statistical approach (random sampling of codes) with the engineering approach (finding optimal codes in genetic and evolutionary algorithms) in the analysis of adaptation of the SGC.

The equation 1 and associated description in Introduction should be moved to Methods.

The assumption on selective values of substitutions in the equation 2 was derived from old studies from 1964 and maybe not valid.

I think that assumption on at least two acidic amino acids (Asp and Glu) and terminal translation signal is also arbitrary. Other assumptions can change the results.

Explain deltaPR in the legend of Fig. 3.

The sentence:” Despite their extremely low probability, the number of such codes was ∼ 10^59.” is repeated.

I did not find the detailed description of genetic algorithms in Methods.

**Have the authors made all data and (if applicable) computational code underlying the findings in their manuscript fully available?**

Reviewer #1: Yes

Reviewer #2: Yes

Reviewer #3: Yes

PLOS authors have the option to publish the peer review history of their article (what does this mean?). If published, this will include your full peer review and any attached files.

Reviewer #1: No

Reviewer #2: No

Reviewer #3: No
---

## [Decision Letter · Decision Letter 1]

16 Mar 2023

Dear Dr Saito,

We are pleased to inform you that your manuscript 'Rare-Event Sampling Analysis Uncovers the Fitness Landscape of the Genetic Code' has been provisionally accepted for publication in PLOS Computational Biology.

Best regards,

Artem Novozhilov

Guest Editor

PLOS Computational Biology

James O'Dwyer

Section Editor

PLOS Computational Biology

I agree with all the reviewers that the authors addressed the reviewer's comments in the revision and in its revised form the manuscript should be accepted for the publication.

Reviewer's Responses to Questions

**Comments to the Authors:**

Reviewer #1: The authors have carefully addressed my previous suggestions.

Reviewer #2: I'm satisfied with the revision.

Reviewer #3: The manuscript is the second version after the review process. I accept the corrections.

**Have the authors made all data and (if applicable) computational code underlying the findings in their manuscript fully available?**

Reviewer #1: Yes

Reviewer #2: Yes

Reviewer #3: **No: **The authors can provide computational code.

PLOS authors have the option to publish the peer review history of their article (what does this mean?). If published, this will include your full peer review and any attached files.

Reviewer #1: No

Reviewer #2: No

Reviewer #3: No

---

## [Editor Report · Acceptance letter]

3 Apr 2023

PCOMPBIOL-D-22-01373R1 

Rare-Event Sampling Analysis Uncovers the Fitness Landscape of the Genetic Code

Dear Dr Saito,

I am pleased to inform you that your manuscript has been formally accepted for publication in PLOS Computational Biology. Your manuscript is now with our production department and you will be notified of the publication date in due course.

With kind regards,

Zsofi Zombor
